# Triploidy in zebrafish larvae: Effects on gene expression, cell size and cell number, growth, development and swimming performance

**Iris L. E. van de Pol**[ID]*, **Gert Flik, Wilco C. E. P. Verberk**[ID]

Department of Animal Ecology and Physiology, Institute for Water and Wetland Research, Radboud University, Nijmegen, The Netherlands

* i.vandepol@science.ru.nl

**Data Availability Statement:** All data files are available from the DANS EASY archive (DOI: https://doi.org/10.17026/dans-xbp-hbxc).

## Abstract

There is renewed interest in the regulation and consequences of cell size adaptations in studies on understanding the ecophysiology of ectotherms. Here we test if induction of triploidy, which increases cell size in zebrafish (*Danio rerio*), makes for a good model system to study consequences of cell size. Ideally, diploid and triploid zebrafish should differ in cell size, but should otherwise be comparable in order to be suitable as a model. We induced triploidy by cold shock and compared diploid and triploid zebrafish larvae under standard rearing conditions for differences in genome size, cell size and cell number, development, growth and swimming performance and expression of housekeeping genes and *hsp70.1*. Triploid zebrafish have larger but fewer cells, and the increase in cell size matched the increase in genome size (+ 50%). Under standard conditions, patterns in gene expression, ontogenetic development and larval growth were near identical between triploids and diploids. However, under demanding conditions (i.e. the maximum swimming velocity during an escape response), triploid larvae performed poorer than their diploid counterparts, especially after repeated stimuli to induce swimming. This result is consistent with the idea that larger cells have less capacity to generate energy, which becomes manifest during repeated physical exertion resulting in increased fatigue. Triploidy induction in zebrafish appears a valid method to increase specifically cell size and this provides a model system to test for consequences of cell size adaptation for the energy budget and swimming performance of this ectothermic vertebrate.

## Introduction

In studies on the ecophysiology of ectotherms, a field gaining more and more interest deals with the regulation and consequences of cell size [1]. For ectotherms, patterns in cell size across thermal clines associated with latitude and altitude have been documented, where animals are generally composed of larger cells in the cold [2–4]. In addition, when ambient temperatures are experimentally lowered while rearing ectotherms, cell size tends to increase and this holds for phyla as diverse as nematodes (*Caenorhabditis elegans*; [5]), arthropods (*Daphnia magna*; [6]) and chordates (the edible frog *Pelophylax esculentus*; [7]). This at least suggests that ectotherms can adaptively change cell size in response to environmental temperature.

**Funding:** Financial Disclosure: This work was supported by The Netherlands Organisation for Scientific Research (W.C.E.P.V., NWO-VIDI grant no. 016.161.321). The funders had no role in study design, data collection and analysis, decision to publish, or preparation of the manuscript.

**Competing interests:** The authors have declared that no competing interests exist.

Understanding the mechanisms and consequences of cell size adaptations can therefore help to better understand the thermal biology of ectotherms. Importantly, cell size may be a determining factor for body size: ectotherms generally grow to a larger body size when reared at low temperatures. This pattern, known as the "temperature-size rule" (TSR), is not fully understood, but more in-depth knowledge of the regulation of cell size may be key to eventually solve this life-history puzzle [1,5,8–10].

An important correlate of cell size is genome size. This correlation between cell size and genome size is generally recognised as fundamental, as it is found in all major animal groups [11–13], yet the underlying mechanisms remain obscure [14]. In general, larger nuclei are required to accommodate a larger genome, and because the nucleocytoplasmic ratio is conserved [11,15], this results in larger cells [15,16]. Interestingly, interspecific differences in DNA content do not predict organismal complexity, the now abandoned C-value paradox [17]. Rather, variation in DNA content reflects varying amounts of non-coding DNA. The possibility thus arises that cell size rather than genome size may be the target of selection [1]. The greatest diversity in genome sizes in vertebrates (and thus cell sizes), is found in fish, by far the most speciose group of the chordate phylum [11].

Before the radiation of teleostean fishes, two rounds of whole genome duplication events occurred in the earliest vertebrates more than 450 million years ago (mya) [18]. This resulted in an ancestral teleost which is predicted to have had 11–13 chromosomes in its haploid set. Around ~ 350 mya another (teleost-specific) whole genome duplication took place [19], explaining why most extant fish have a haploid chromosome number of 24 or 25 [20,21]. In both the salmonid lineage and in part of the cyprinid lineage further polyploidization occurred (examples are the tetraploid common carp and goldfish, which possess a total number of 100 chromosomes). Also, in the ancestral lineage of *Acipenceriformes*, repeated rounds of whole genome duplication events lead to an impressive hexaploid genome with approximately 368 chromosomes in some extant sturgeons (*Acipenser baerii*) [22,23].

The consecutive whole genome duplications did not result in an exponential increase in genome size (C-value) in fish: increases in genome size must have been balanced by subsequent genome size reduction, possibly because cell size is under stabilizing selection [24]. Zebrafish (*Danio rerio*) are *Cyprinidae*, but did not undergo the whole genome duplication seen in other cyprinids and have 48 chromosomes. While duplicates of several genes are found in the zebrafish genome (see for example [25]), these are derived from the teleost specific whole genome duplication. As the teleost specific whole genome duplication was followed by gene (sub-) functionalisation and genome reduction, zebrafish are considered to be diploid animals, setting them apart from tetraploid cyprinids [26].

Cell size has direct consequences for energy metabolism: larger cells have lower metabolic rates on a per mass basis [15]. One of the main reasons for this is the lower membrane surface area relative to cellular volume in larger cells. A larger cell has lower energetic costs for maintaining electrochemical gradients across its plasma membrane, a process that may take up to 30% of the total energy budget of the cell [27]. In other words, large cells are more energy efficient with a smaller plasma membrane compartment to sustain. The flip side is that large cells have a lower capacity for transmembrane transport of resources such as nutrients and oxygen, and longer intracellular diffusion distances, potentially making them more susceptible to oxygen limitation. In this respect, smaller cells could provide the advantage of greater performance. Thus, cell size affects the balance between resource uptake and costs for maintaining ionic gradients [28–30].

Ploidy can be artificially increased in zebrafish, either by inducing triploidy [31–33], or tetraploidy, although tetraploid zebrafish do not to survive beyond 50 days of age [34]. This is surprising in the light of the tetraploidy of the closely related common carp (*Cyprinus carpio*).

Triploid zebrafish, however, survive easily well up to adult age (> 1 year, personal observations), although they all develop into males [35]. Artificial triploidy induction is a common practice in aquaculture, usually performed to achieve sterility in populations of fish, mainly salmonids [36]. Benefits of anthropogenic triploidy are prevention of precocious maturation, which may lead to a larger body size as there is no major investment in gametes (mature ovaries may take a large volume of the body weight and is often not commercially appreciated), as well as avoidance of genome fouling of wild populations when cultured fish escape and hybridize with natural populations [37]. However, triploid fish have been reported to be more susceptible to stressors such as high temperatures and hypoxia [38,39]. It has been suggested that this susceptibility is mainly caused by issues with oxygen delivery [40]; indeed, a combination of both high temperatures and hypoxia significantly increased mortality in triploid Atlantic salmon, compared to solely high temperatures [41].

Here we report on triploidy induction in zebrafish to enlarge its cells to study the consequences of cell size adaptations for whole-organism performance. For this model to function properly, triploid zebrafish should ideally only differ from wildtype diploids in cell size, when reared under standard conditions. We compared diploid and triploid zebrafish larvae at three levels of biological organisation, *i.e.* the genomic, single cell and whole-organism level. Triploid zebrafish have been used for studies in fields as diverse as toxicology and theriogenology [31–33,35], but to the best of our knowledge this is a novel, more in-depth comparison between diploid and triploid zebrafish focused on the characterisation of triploid zebrafish in the embryonic and larval stage. We hypothesised and confirm that under non- demanding (*i.e.* standard rearing) conditions, diploid and triploid zebrafish larvae are highly comparable in expression levels of housekeeping genes, growth, development and morphology. In an escape response trial, where maximum performance of the fish is required, we predict that triploids will be outperformed by diploids based on a lower capacity of larger cells to generate energy.

## Materials and methods

### Zebrafish husbandry

Breeding stock of adult zebrafish from the AB strain (wild-type strain supplied by ZIRC, ZFIN ID: ZDB-GENO-960809-7, bred for maximum three successive generations at the Radboud Zebrafish Facility, Nijmegen, The Netherlands) were kept in 4-L tanks with recirculating tap water (pH 7.5 –pH 8) at a density of approximately 30 fish per tank. Zebrafish stock was fed twice a day with Gemma Micro 300 Zf (5% of bodyweight per day; Nutreco N.V., Amersfoort, The Netherlands), with addition of *Artemia* once a day, and were maintained at a continuous cycle of 14 hours light and 10 hours darkness at a temperature of 27˚C.

All experiments were carried out in accordance with the Dutch Animals Act (https://wetten.overheid.nl/BWBR0003081/2019-01-01), the European guidelines for animal experiments (Directive 2010/63/EU; https://eur-lex.europa.eu/legal-content/EN/TXT/?uri=CELEX:32010L0063) and institutional regulations.

### Egg collection and triploidy induction

The obtained eggs were fertilised *in vitro* to accurately determine the timing of fertilisation, which is crucial for triploidy induction. Males and females were separated the day before *in vitro* fertilisation (ivf). The following morning, ivf was performed as described in the protocol of 'The Zebrafish Book' [42]. Males were anaesthetised in 0.05% v/v 2- phenoxyethanol and blotted damp-dry. Sperm was collected by use of a P-10 pipette fitted with a plastic micro-10 tip (Greiner Bio-One, Kremsmünster, Austria), and applying gentle suction at the opening of the cloaca while stroking the sides of the fish. The quality of the sperm was assessed by colour;

sperm was only used when its appearance was milky white and not watery. The sperm of 8–10 males was pooled in isotonic Hank's saline solution (137 mM NaCl, 5.4 mM KCl, 0.25 mM Na$_2$HPO$_4$, 0.44 mM KH$_2$PO$_4$, 1.3 mM CaCl$_2$, 1.0 mM MgSO$_4$, 4.2 mM NaHCO$_3$) using 50 μL to dilute sperm from one male.

The pooled sperm sample was kept on ice while proceeding with females. They anaesthetised similarly to the males, but were placed in a petri dish (with no water) to collect the eggs following gentle pressure on the belly. The quality of the eggs was assessed by colour and shape; we only used eggs that appeared yellowish and translucent, with a regular round shape. Immediately after egg collection, 50 μL of the sperm solution was added followed by the addition of E2 embryo medium (5 mM NaCl, 0.17 mM KCl, 0.33 mM CaCl$_2$, 0.33 mM MgSO$_4$). Meiosis II is initiated when eggs come into contact with a hypotonic solution such as water or E2 medium. To maximize fertilisation, first we enveloped only the egg mass with E2 medium, and after 30 seconds the petri dish was filled up.

A cold shock to inhibit meiosis II was applied 3 minutes after the addition of E2 to induce triploidy, as the second polar body is then retained within the nucleus of the diploid egg. The cold shock was given by incubation of the eggs (contained in a plastic tea strainer) in a water bath at 4˚C for 20 minutes; next the eggs were allowed to warm up in a water bath at 28˚C for 5 minutes, and then transferred to the experimental setup. Starting with good quality eggs and sperm, shock temperature, duration of the exposure and timing post fertilisation are the critical variables that influence the efficiency of triploidy induction. To ensure a high induction efficiency, we tested a variety of conditions, listed in S1 Table. We initially used heat as a temperature shock to induce triploidy in zebrafish as described by Kavumpurath and Pandian (1990) [32]; however, we obtained better and more consistent results with a cold shock. This seems conflicting with the results of others [31,43], although in these studies slightly different conditions were applied. Especially timing post fertilisation seems to be a major factor affecting survival rates and triploidy efficiency; Franek et al., (2019) [31] tested with initiation of the temperature shock up to 2 minutes post fertilisation and Hou et al., (2015) [43] immediately (within 10 seconds) applied a cold shock to the fertilised eggs, whereas we achieved optimal results with applying the cold shock 3 minutes post fertilisation, when the eggs have sufficiently swelled in room temperature E2 medium.

Throughout the manuscript, embryos that received a cold shock to induce triploidy are referred to as cold shocked embryos (as triploidy could only be verified at the end of the experiments, we chose not to refer *a priori* to these individuals as triploids).

### Larval maintenance

Within an hour after fertilisation, diploid and cold shocked embryos were transferred to a 48-wells plate (three embryos per well; Greiner Bio-One, Kremsmünster, Austria) with a mesh bottom, permeable to water, placed in a breeding tank with E3 medium (E2 medium with addition of 10–5% methylene blue). The breeding tank was transferred as a whole to a water bath to maintain a constant temperature of 26.5˚C (+/- 0.2˚C) and constant aeration of E3 medium was provided (setup based on Khaliullina-Skultety et al. (2017) [44]). Larvae were maintained up until 5 dpf.

### Growth and development

Embryos were staged at fixed time points according to Kimmel et al. (1995) [45] using a Leica MZ FLIII stereomicroscope (Leica microsystems, Wetzlar, Germany). Staging was based on clearly distinguishable features, *i.e.* 6 hours post fertilization (hpf, embryonic shield), 24 hpf (heart beat and early pigmentation), 30 hpf (weak circulation), 48 hpf (tapering yolk extension)

and 72 hpf (protruding mouth). At each time point 5 embryos were staged from the diploid and cold shocked groups. At 5 dpf, length in mm (to the nearest 0.01 mm) was assessed from pictures of living larvae taken with a Leica MZ FLIII stereomicroscope and the segmented line tool in the ImageJ program (https://imagej.nih.gov/ij/).

## qPCR

At 24, 48, 72, 96 and 120 hpf embryos and larvae were collected in 2 mL Eppendorf tubes and instantly frozen in liquid nitrogen for qPCR analysis. At each time point, three individuals were pooled in one sample and three replicate samples (a total of 9 larvae) were obtained from both the diploid and cold shocked groups. All samples were stored at -80°C until further use.

Total RNA was isolated with TRIzol (Thermo Fisher Scientific, Waltham, MA, USA) according to the manufacturer's instruction with minor changes. Samples were homogenized in 400 μL TRIzol with a 3 mm glass grinding bead using a mixer mill (Thermo Fisher Scientific, Waltham, MA, USA) for 30 seconds at 20 Hz. An extra ethanol precipitation was performed and RNA pellets were dissolved in 15 μL diethyl pyrocarbonate (DEPC) $H_2O$.

To obtain equal amounts of RNA, concentrations were measured by nanodrop spectrophotometry (ND-1000; Isogen Life Science B.V., De Meern, The Netherlands). For DNase I (Thermo Fisher Scientific, Waltham, MA, USA) treatment and cDNA synthesis 500 ng total RNA was used. Synthesis of cDNA was carried out with Superscript Reverse Transcriptase II enzyme (Thermo Fisher Scientific, Waltham, MA, USA) according to manufacturer's instructions. The obtained cDNA was diluted 10x in DEPC $H_2O$ for qPCR. A total PCR volume of 20 μL was used, including 4 μL cDNA and 10 μL iQ SYBR Green Supermix (Bio-rad, Hercules, CA, USA). Cycling parameters were: 3 minutes at 95°C, 40 PCR cycles of 15 seconds at 90°C and 60 seconds at 60°C and finally a melt curve procedure from 65°C to 95°C (ΔT +0.5°C/ cycle).

In total the expression levels of six housekeeping genes were analysed: ribosomal protein L13a (*rpl13a*), eukaryotic translation elongation factor 1 alpha1, like 1 (*eef1a1l1*), actin, beta 1 (*actb1*), polymerase (RNA) II (DNA directed) polypeptide D (*polr2d*), ribosomal protein S11 (*rps11*, previously known as *40S*) and TATA box binding protein (*tbp*). These genes are involved in a range of basic cellular processes and therefore frequently used to normalize gene expression levels of other genes (Table 1). To ensure a non-biased interpretation of the expression values, the relative normalized expression for each housekeeping gene was calculated using a combined index of the relative quantity from the other five housekeeping genes [46,47]. In addition, the expression of heat shock protein 70 (*hsp70.1*) was analysed, as an indicator of potential thermally induced stress in triploids [48,49] (Table 1).

## Genome and cell size

At 5 dpf, cold shocked larvae were collected to verify triploidy induction by analysis of the amount of nuclear DNA. An individual cold shocked larva was pooled with a diploid control of the same age to serve as an internal control. For qPCR it was not possible to analyse ploidy level of individuals, because whole larvae were used for RNA isolation. In this case, other larvae from the same batch were used to calculate the efficiency of triploidy induction.

One cold shocked and one diploid control larva were pooled in 1.5 mL Eppendorf tubes, chilled ice-cooled E3 medium was added and the larvae were left on ice for 20 minutes to be euthanized. After removing E3 medium, larvae were pre-treated with 150 μL lysis buffer (0.1% sodium citrate, 0.1% Triton X-100 in PBS) and they were immediately frozen in buffer at -20°C until further processing (storage up to two weeks did not affect downstream processing).

**Table 1. Primer sequences for qPCR.**

| Gene | Source | Process | Fw primer sequence (5'-3') | Rv primer sequence (5'-3') |
|------|--------|---------|---------------------------|---------------------------|
| *rpl13a* | NM_212784.1 | Translation | TCTGGAGGACTGTAAGAGGTATGC | AGACGCACAATCTTGAGAGCAG |
| *eef1a1l1* | NM_131263.1 | Cell cycle | CTGGAGGCCAGCTCAAACAT | TCAAGAAGAGTAGTACCGCTAGCATTAC |
| *actb1* | NM_131031.1 | Cytoskeleton integrity | CTTGCTCCTTCCACCATGAA | CTGCTTGCTGATCCACATCT |
| *polr2d* | NM_001002317.2 | Transcription | CCAGATTCAGCCGCTTCAAG | CAAACTGGGAATGAGGGCTT |
| *rps11* | NM_213377.1 | Translation | GCTTCAAAACCCCCAGAGAA | TCAGGACGTTGAACCTCACA |
| *tbp* | NM_200096.1 | Transcription | CTTACCCACCAGCAGTTTAGCAG | CCTTGGCACCTGTGAGTACGACTTTG |
| *hsp70.1* | NM_001362359.1 | Cellular stress response | GACATCGACGCCAACGGG | GCAGAAATCTTCTCTCTCTGC |

Samples were thawed and placed on ice before mechanical dissociation. Larvae were slowly passed through a 25Gx1"/0.5x25 mm needle fitted to a 1 mL syringe (Henke Sass Wolf, Tuttlingen), which was repeated 10 times. The homogenate was filtered with a 70 μm mesh cell strainer (pluriSelect Life Science, Leipzig, Germany) topped on a 1.5 mL Eppendorf tube to obtain single nuclei. After allowing the nuclei to flow through for 10 minutes, samples were centrifuged for 4 minutes at 1000 rpm at 4°C. The supernatant was carefully removed, leaving a small drop of liquid on the invisible pellet. Finally, samples were incubated overnight with 300 μL freshly prepared propidium iodide (PI) staining buffer (0.1% w/v sodium citrate solution, 20 μg/ml PI (Merck KGaA, Darmstadt, Germany), 0.1 mg/ml RNAse A (Thermo Fisher Scientific, Waltham, MA, USA) and 0.1% Triton X –100), keeping the samples on ice and in the fridge shielded from light. The next morning, samples were transferred to 5 mL test tubes and kept on ice until analysis with a FC500 5-color Flow Cytometer (Beckman Coulter Life Science, Indianapolis, IN, USA).

To verify triploidy, the R package {flowPloidy} by Smith et al. (2018) [50] was used. All samples consist of two pooled larvae; one diploid larva that serves as an internal control and one cold shocked larva for which ploidy status is to be established. Including an internal control with known genome size is a standard practice to relate fluorescent intensities of peaks in the obtained DNA histograms to the corresponding nuclear DNA content [51,52]. In Fig 1, two exemplary DNA histograms are shown for triploidy identification. The first peak represents G1 phase cells of the diploid control larva. This peak is annotated as the standard peak with a genome size of 1.44 pg, the genome size of zebrafish [53]. These cells have a 2n DNA content. When both larvae are diploid (Fig 1A), there are two main peaks; in addition to the G1 phase cells there is a second peak showing G2 phase cells with a 4n DNA content. When the sample included a triploid cold shocked larva in addition to the internal control diploid larvae, we observe G1 phase cells with a 2n and a 3n DNA content (the first two peaks), with the peak for triploid larvae being displaced such that the genome size corresponds to a 50% increase. A similar shift is observed for the G2 phase cells. Thus, the fourth peak shows G2 phase cells of the triploid larva, which have a 6n DNA content. Using the annotation tools of the {flowPloidy} R package, a model was fitted over the raw FCM data. FlowPloidy analysis incorporates the continuous aggregate model to correct for the presence of doublets, triplets and quadruplets in 4n, 6n and 8n peaks, and a residual Chi-Square (RCS) value was calculated, which is a rough goodness-of-fit value [54].

The obtained DNA histograms were used to calculate the number of cells of diploid and triploid larvae in pooled samples. These are not the absolute cell numbers that compose a larva, but as we consistently used a diploid larva as internal control we can calculate the relative difference in cell numbers between diploids and triploids. We discriminated between G2 phase cells of triploid larvae and endopolyploid cells of diploid larvae which have an overlapping peak at 6n, by calculating the proportion of 6n cells derived from diploid larvae based on

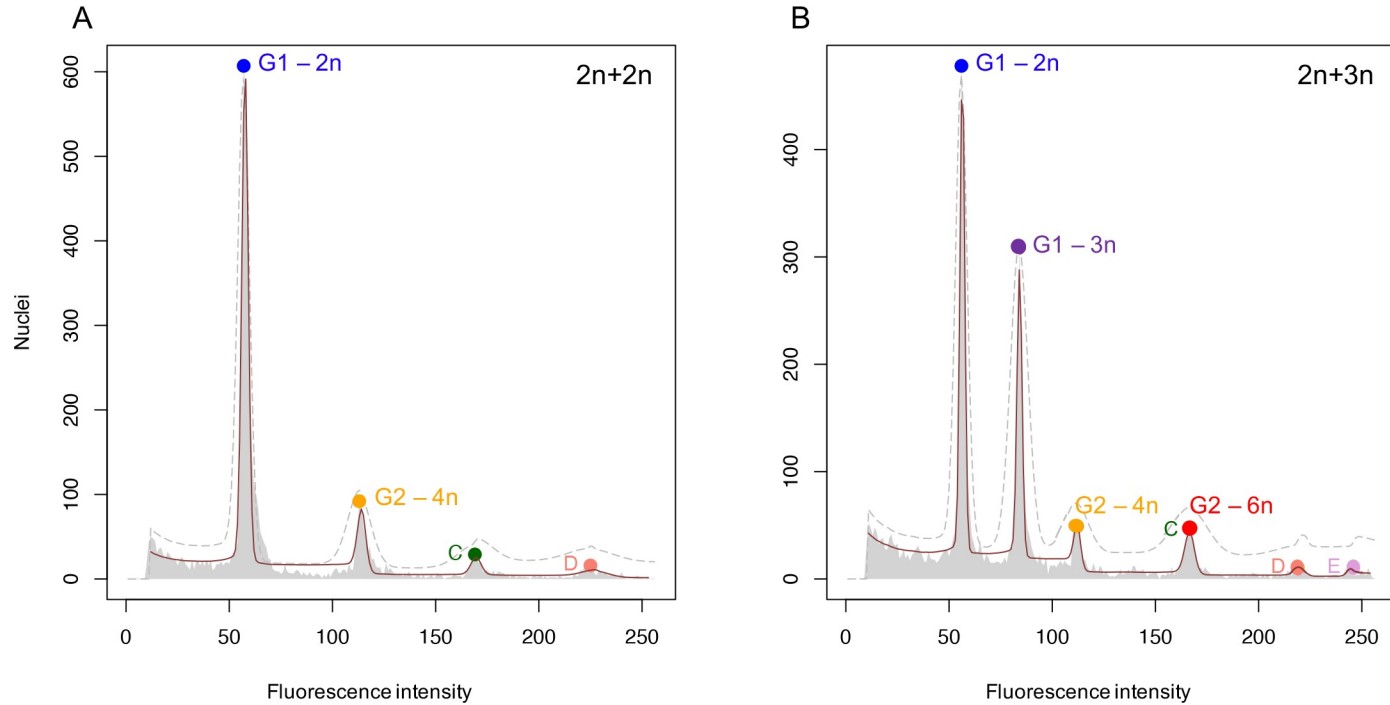

**Fig 1. Assessment of triploidy induction.** A) Exemplary DNA histogram of two diploid larvae pooled in one sample. Standard size G1-peak = 1.44 pg, estimated size G2-peak = 2.85 pg, ratio G2 / G1 = 1.98, RCS value = 6.96. B) Exemplary DNA histogram of one diploid and one triploid larva pooled in one sample. Standard size G1-peak 2n = 1.44 pg, estimated size G1-peak 3n = 2.15 pg, ratio G1-3n / G1-2n = 1.49, RCS value = 2.11. In both A) and B) the dashed grey line is the initial model estimate and the red line is the fitted model. Colored circles represent matching cell cycle phases derived from a diploid or triploid larva. Peaks C, D, and E represent endopolyploid cell populations.

the proportion of 6n cells in diploid control samples. We also measured the numbers of cells in the G1 and G2 phase (presented as ratio G2/G1); this ratio is an indicator of the dividing potential of the cells.

Cell size was analysed in erythrocytes of adult (>1.5 years old) diploid and triploid zebrafish. Blood was obtained from zebrafish euthanized with an overdose of 0.1% v/v 2-phenoxyethanol and a standard H&E staining was performed on blood smears. Pictures were taken with a Leica DM RBE microscope and cell area was measured by using automatic particle analysis of the program ImageJ (https://imagej.nih.gov/ij/). In each picture the cell area was obtained for 20 cells. We took care to only obtain area measurements for cells that were not touching other cells. Cell areas were converted to cell volumes as follows:

$$\text{Cell volume} = \text{Cell area}^{3/2}$$

## Swimming performance

Maximum swimming velocity of diploid and cold shocked larvae was assessed using a Danio-Vision system (Noldus Information Technology B.V., Wageningen, The Netherlands). At 5 dpf, larvae were transferred to a 24-wells (Greiner Bio-One, Kremsmünster, Austria) plate containing 1 mL of E3 medium in each well. Care was taken to avoid using larvae with visual morphological abnormalities (i.e. a curved body axis, pericardial oedema and underdeveloped head and eyes). The 24-well plates were filled with a combination of cold shocked and diploid larvae. They were placed individually in a well and were acclimated for 10 minutes in the setup at 26.5°C before the start of the experiment. All experiments were performed in the afternoon,

as activity levels of larvae are most stable during this time of day [55]. Larvae were subjected to a startle protocol as described in van den Bos et al. (2017) [56]. After 10 minutes, 10 tap stimuli were presented with an interval of 20 seconds. This protocol was chosen as previous studies showed no habituation in the startle response of zebrafish larvae to repeated stimuli with a 20 seconds inter-stimulus interval [57–59]. The maximum velocity during the startle response was measured in mm/s. Subsequent analysis were performed only with larvae that at least once exhibited a response higher then 15 mm/s.

## Statistical analyses

All analyses were performed using RStudio version 1.1.383; a significance threshold of $\alpha = 0.05$ was respected. Length measurements for diploid and triploid larvae were compared using a two sample $t$-test, as an $F$-test confirmed equal variances. Developmental rates of diploids and triploids were compared using a general linear model, for which we visually checked the frequency distribution of the residuals as being normally distributed. With a subsequent ANOVA we tested for the effects of ploidy and hpf on developmental stages and the interaction thereof. Gene expression data was analysed for each gene separately, using a general linear model (again the residuals were visually checked for normality) and subsequent ANOVA to test for differences between ploidy levels and the interaction with dpf. Post hoc Tukey's test was performed to find differences between ploidies at specific days. To compare cell numbers between diploid and triploid larvae we used a paired $t$-test, as each sample pooled a diploid and a triploid larva. For comparing the ratio of cells in the G1 and G2 phase between diploids and triploids, we used Welch's two sample t-test, as an $F$- test revealed unequal variances. The estimates of cell volumes of diploid and triploid erythrocytes were compared using a two sample $t$-test, as an $F$-test confirmed equal variances. For analysis of the swimming performance data, we only included responsive larvae (they at least once showed a response higher than 15 mm/s). Pearson's $\chi^2$-test revealed that the proportion of non-responsive larvae was higher in triploids ($\chi^2 = 16.41$, $df = 1$, $p < 0.001$, see also Table 2). We tested for the effect of ploidy level on maximum swimming velocity separately for startle 1, startle 2 and startle 3–10 with ANOVA, using a linear mixed effects model [60] which included trial as a random factor. In addition, given the bimodality in responses observed, we compared the proportion of larvae that responded to a stimulus separately for startle 1, startle 2 and startle 3–10 using Pearson's $\chi^2$-tests. From those measurements where larvae surpassed the threshold swimming speed of 15 mm/s, we also compared the maximum swimming velocity between diploids and triploids using a two sample $t$-test, as an $F$-test confirmed equal variances.

## Results

### Triploidy induction and cellular architecture

The efficiency of triploidy induction was on average just over 98% (S1 Table). Survival rates at 24 hpf did not significantly differ between cold shocked and control embryos (70.8% ± 5.4 s.d. and 77.2% ± 5.8 s.d., respectively; $df = 7$, $p = 0.13$). Ploidy level could be reliably verified with flow cytometry, by assessing the fluorescence intensity of the second peak in the DNA

**Table 2. Numbers of responsive and non-responsive larvae in the startle protocol.**

| Ploidy level \ Response type | Responsive | Non-responsive | Total |
|---|---|---|---|
| # 2n larvae | 71 | 2 | 73 |
| # 3n larvae | 53 | 21 | 74 |
| Total | 124 | 23 | |

histogram compared to the standard peak of the internal diploid control (Fig 1). Triploid larvae showed a 50.8% (± 0.9 s.d., $n$ = 51) increase in DNA content, matching the expected increase in genome size by a factor of 1.5. Strikingly, the DNA histograms for diploid and triploid samples showed, apart from the peaks representing the G1 and G2 phase, other small peaks with polyploid 6n and 8n cells for diploid larvae and 9n cells for triploid larvae (Fig 1).

As we used a diploid larva as an internal control, we could also compare relative cell counts between diploid and triploid larvae on a per sample basis, showing that on average the cell count is 1.72 (± 0.70 s.d.) times lower in triploids compared to diploids (Fig 2A, $df$ = 49, $p < 0.001$). The ratio of G2 to G1 phase cells is larger in triploid larvae (Fig 2B, $df$ = 82.025, $p < 0.001$), indicating that more cells are in the process of dividing in triploids than in diploids. As diploid and triploid zebrafish larvae did not differ in length (see below, Fig 6), triploid larvae can be inferred to consist of larger but fewer cells, compared to their diploid counterparts. The 1.72 (± 0.70 s.d.) times lower cell count in triploids agrees well with the expected 1.5 times increase in cell size, as genome size increased by the same factor [11]. We confirmed the increase in cell size in erythrocytes from adult diploid and triploid zebrafish, demonstrating that the estimated cell volumes of triploid erythrocytes are on average 1.64 (± 0.18 s.d.) times larger than diploid erythrocytes (Fig 3).

### Expression of housekeeping genes

Expression levels of the housekeeping genes *rps11*, *actb1* and *eef1a1l1* were the same for diploid and triploid larvae during development (Fig 4A, 4B and 4C). For the other three genes, the ontogenetic changes in gene expression levels followed the same trajectory, although here expression levels of *rpl13a* were slightly lower in triploid larvae compared to diploids (Fig 4E, $F_{1,20}$ = 7.00, $p$ = 0.016), whereas *tbp* expression was slightly higher in triploid larvae (Fig 4F,

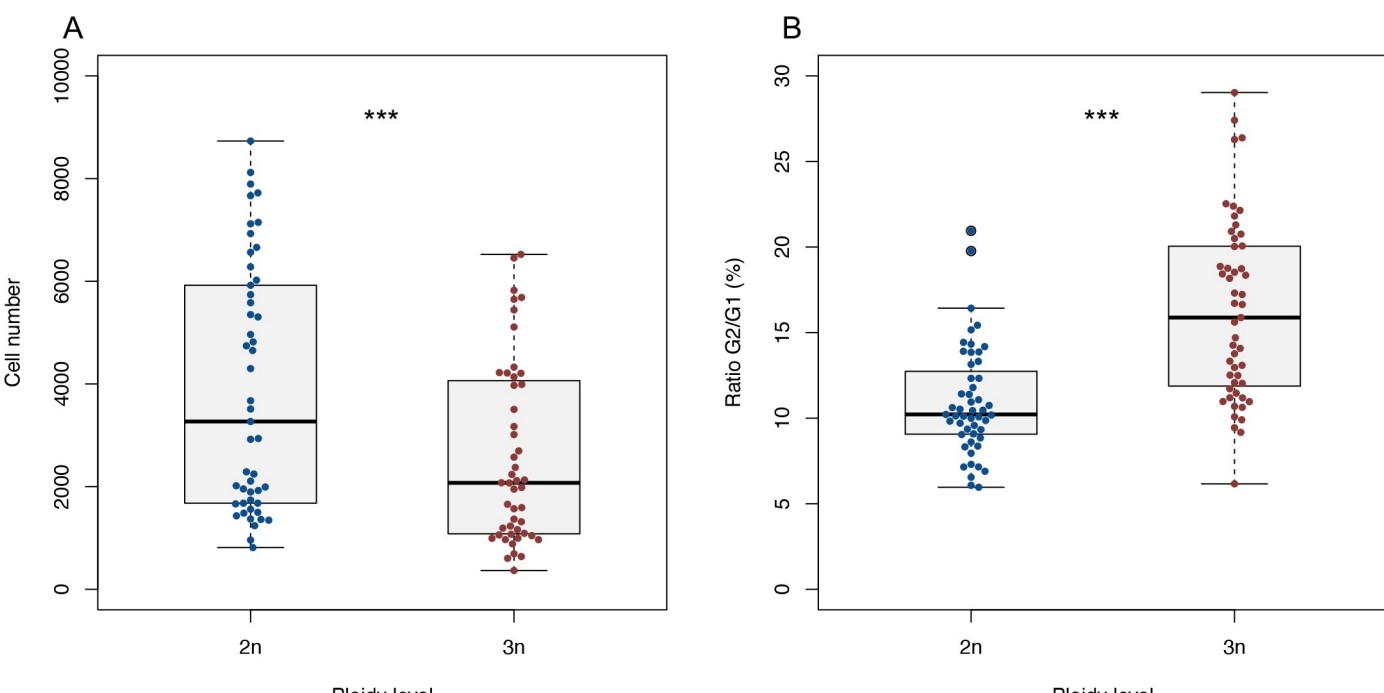

**Fig 2. Cell count and G2/G1 ratio of diploid and triploid larvae.** A) Cell number of diploid and triploid larvae in pooled FCM samples. Paired *t*-test, *** $p < 0.001$, $n$ = 50. B) Ratio of cells in G1 and G2 phase, calculated as G2/G1*100 for each diploid and triploid larva. Welch's *t*-test, *** $p < 0.001$, $n$ = 50.

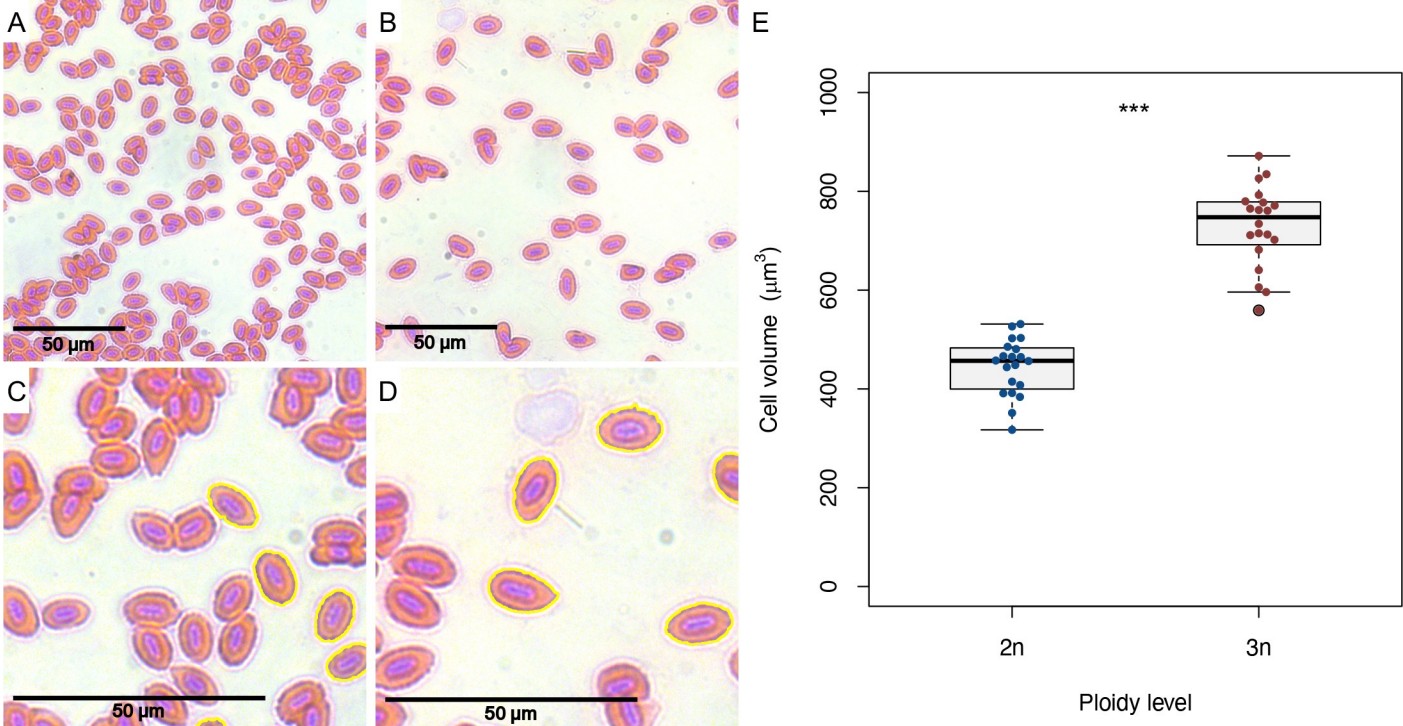

**Fig 3. Diploid and triploid erythrocytes and estimated cell volumes.** A) Microscopic picture of diploid erythrocytes (40x). B) Microscopic picture of triploid erythrocytes (40x). C) Zoomed in version of picture A, showing automatic cell area detection of diploid cells. D) Zoomed in version of picture B, showing automatic cell area detection of triploid cells. Note that only non-touching cells are analysed. E) Estimated cell volumes of diploid and triploid erythrocytes. Two sample *t*-test, *** $p < 0.001$, $n = 40$.

$F_{1,20} = 5.36$, $p = 0.031$). Expression of *polr2d* through time varied with ploidy level (the interaction between ploidy and dpf was significant: Fig 4D, $F_{4,20} = 3.72$, $p = 0.020$). A Tukey's post-hoc test revealed that *polr2d* gene expression was significantly higher at 1 dpf ($p = 0.038$) and 2 dpf ($p = 0.037$) in diploid larvae, but later in development the expression values of diploids and triploids converged (Fig 4D). There was no significant difference in the expression levels of *hsp70.1* between diploid and triploid larvae (Fig 5, $F_{1,20} = 0.77$, $p = 0.39$). Note that *hsp70.1* expression levels were normalized using the housekeeping genes *eef1a1l1* and *actb1*, as these genes are expressed most stably during development and they are similarly expressed in diploids and triploids.

## Growth and development

Diploid and triploid zebrafish larvae are morphologically indistinguishable at 5 dpf (Fig 6B and 6C). The average length for diploids is 3.97 mm (± 0.15 s.d., $n = 66$) and for triploids 3.97 mm (± 0.19 s.d., $n = 62$) (Fig 6A; t$_{1, 126}$ = -0.051, $p = 0.959$). In addition, the timing of developmental landmarks did not change with ploidy (Fig 7, $F_{1,133} = 0.519$, $p = 0.473$). There were two larvae within the triploid group that lagged behind in their development and could therefore be considered outliers; however, analyses that excluded them produced the same results.

## Swimming performance

During the startle protocol, most larvae increased swimming velocity at least once (responsive larvae), although some did never (non-responsive larvae). Within the responsive larvae (124 out of 147 individuals), maximum swimming velocity of triploid zebrafish larvae was lower

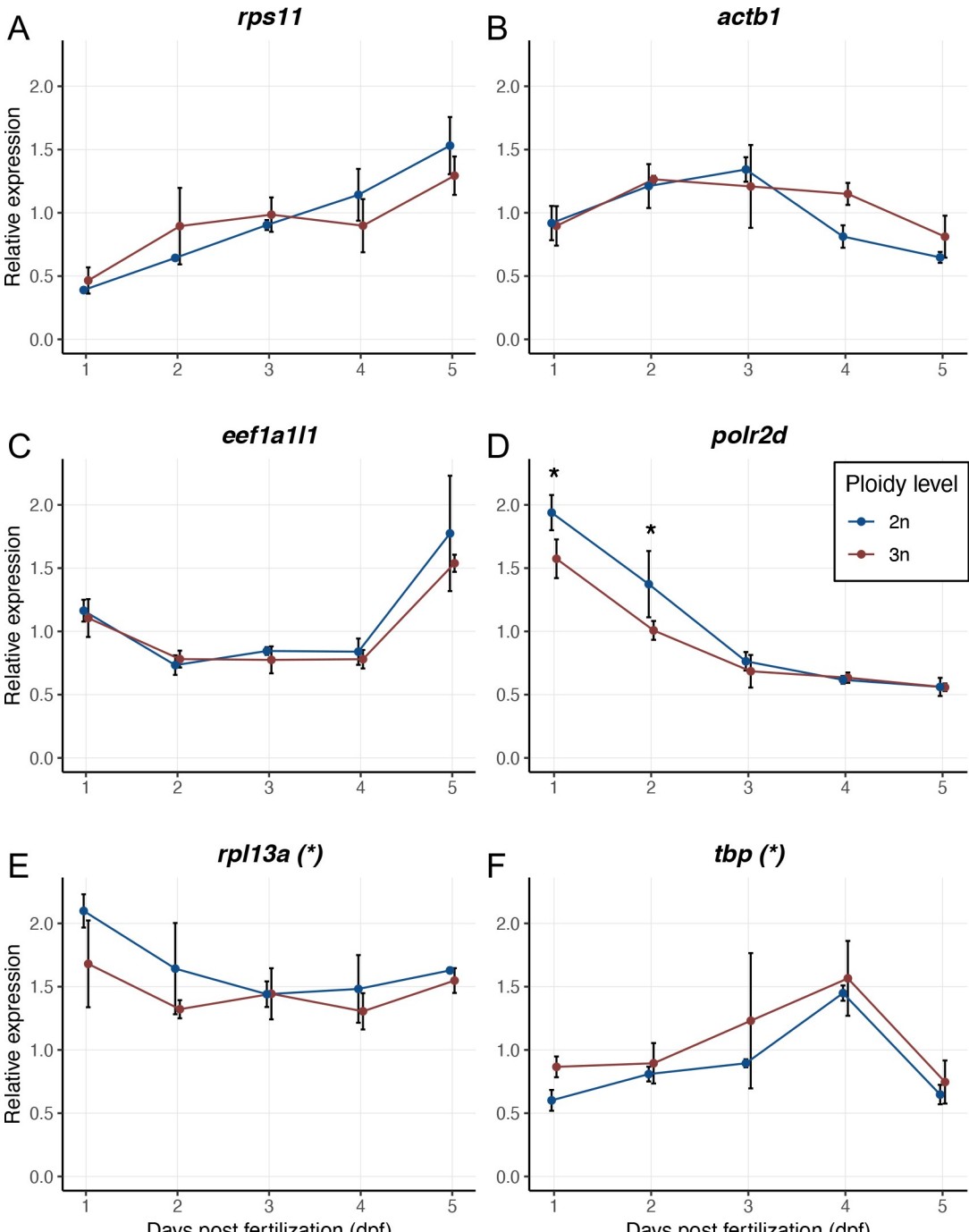

**Fig 4. Relative expression values of housekeeping genes in diploid and triploid larvae during development.** A) *rps11*, ribosomal protein S11. B) *actb1*, actin, beta 1. C) *eef1a1l1*, eukaryotic translation elongation factor 1 alpha 1, like 1. D) *polr2d*, polymerase (RNA) II (DNA directed) polypeptide D. E) *rpl13a*, ribosomal protein L13a. F) *tbp*, TATA box binding protein. Expression values for each gene are normalized using a combined index of the relative quantity of the other five housekeeping genes. Values are presented as means with standard deviations. Asterisks in titles indicate a significant difference in expression levels without an interaction with dpf. ANOVA, * $p < 0.05$, $n$ = 3x3 larvae per ploidy level per day.

relative to diploids at the first startle stimulus (Fig 8A, $F_{1,122} = 12.1$, $p < 0.001$), and further decreased at subsequent stimuli (Fig 8B and 8C; $F_{1,122} = 15.0$, $p < 0.001$ and $F_{1,122} = 31.4$, $p < 0.001$, respectively). There was a clear bimodality in the swimming velocity: either larvae

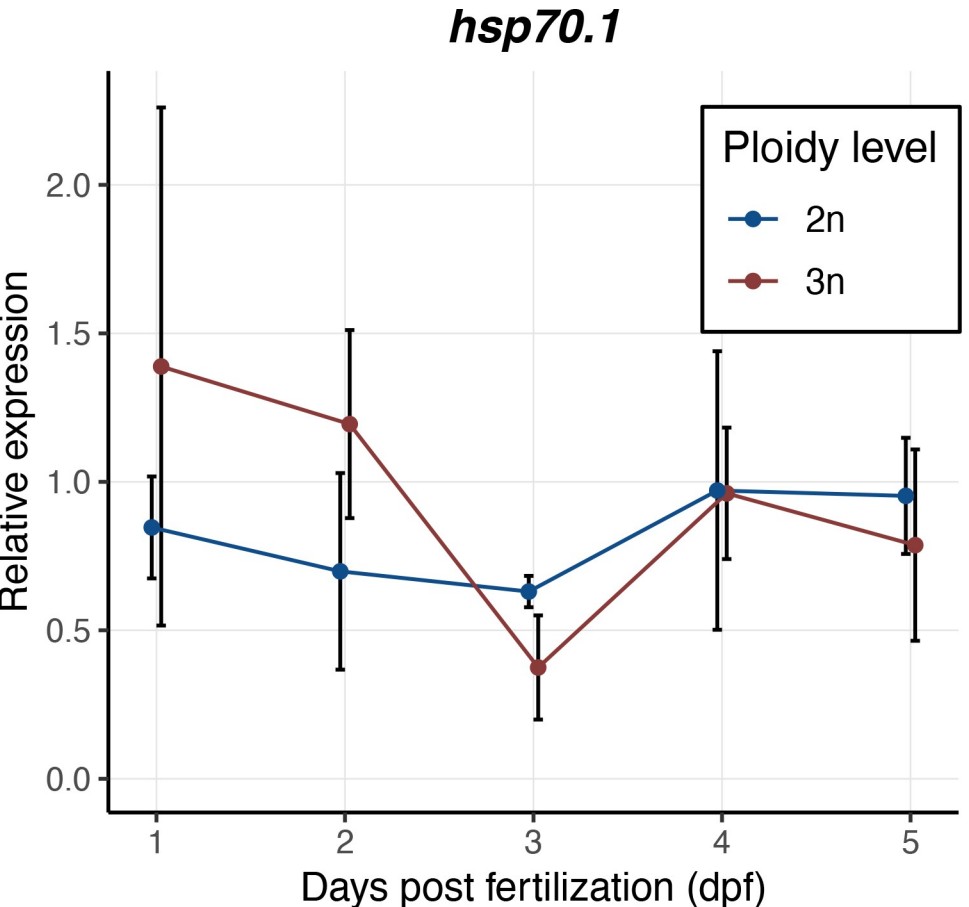

**Fig 5. Relative expression values of *hsp70.1* in diploid and triploid larvae during development.** Expression values are normalized using a combined index of the relative quantity of the housekeeping genes *eef1a1l1* and *actb1*. Values are presented as means with standard deviations. ANOVA, *p* = 0.39, *n* = 3x3 larvae per ploidy level per day.

reacted or did not and the threshold value demarcating the two groups was at 15 mm/s. When only considering the swimming velocities above this threshold value, there was no significant difference in the average maximum velocity between diploids and triploids (startle 1: *t* = 0.54, *df* = 108, *p* = 0.59, startle 2: *t* = -1.1624, *df* = 77, *p* = 0.25, startle 3–10: *t* = 0.93, *df* = 414, *p* = 0.35). Thus, the decline in swimming velocity in triploids with repeated trials is driven by triploids becoming increasingly unresponsive (for analysis see S1 Appendix).

## Discussion

To investigate how differences in cell size affect performance at the whole-organism level, a comparison is required between individuals that differ in cell size only, being similar in other respects. Our results demonstrate that triploid zebrafish could be a suitable model system. Triploidy in zebrafish leads to larvae with larger, but fewer cells than in their diploid counter-parts. DNA content *increased* by a factor of 1.5, with concomitant effects on nuclear size and cell size [1,11,14,61], whereas cell number *decreased* by roughly the same factor. The increase in cell size following triploidy induction was confirmed in erythrocytes from adult diploid and triploid zebrafish. The cold shock protocol established for this research reliably induces trip-loidy (>98%). Also, mortality rates of triploids were not different from diploid controls (S1 Table). Possible timing irregularities could explain our variable results when applying heat

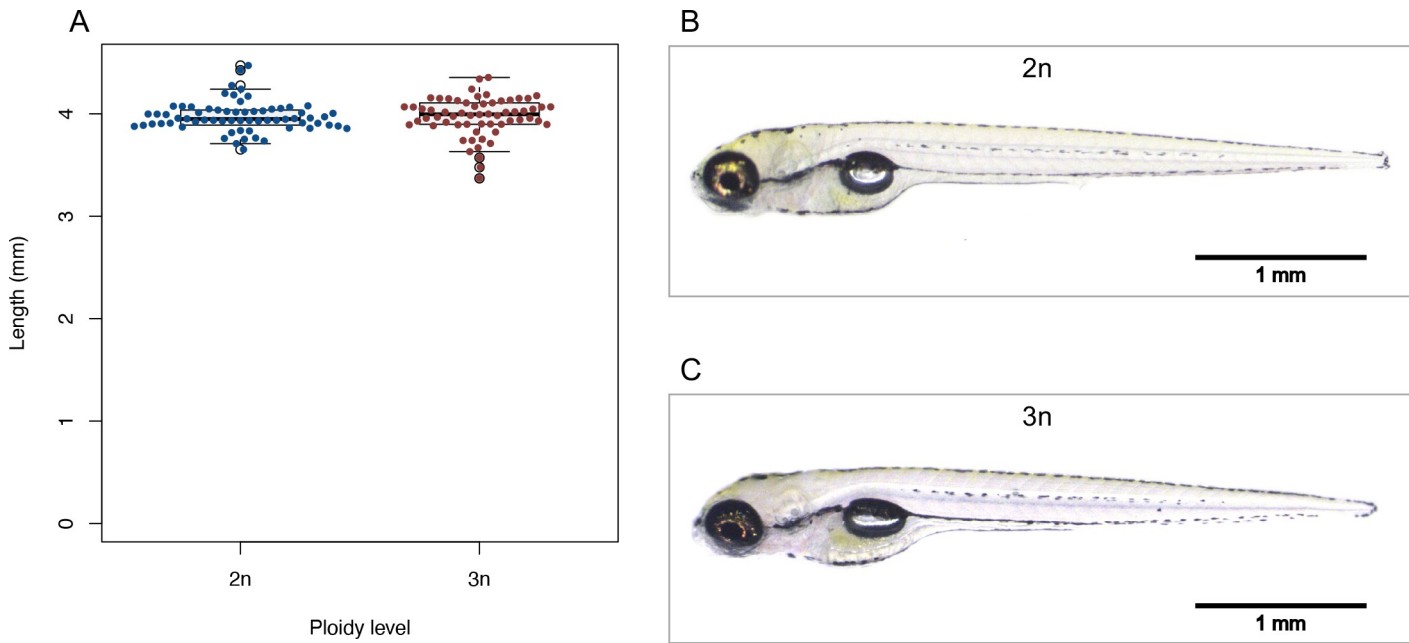

**Fig 6. Body length and morphology of diploid and triploid larvae.** A) Body length of diploid and triploid larvae. Two sample *t*-test, $p = 0.96$, $n = 128$. B) Representative picture of diploid larva at 5 dpf. C) Representative picture of triploid larva at 5 dpf.

shocks (41˚C), as initially the timing of fertilisation was determined by visually assessing egg release from females during natural fertilisation. As females do not always release all eggs simultaneously, timing irregularities up to 10 seconds could occur. These small differences in timing could have a larger effect during heat shocks than during cold shocks, as the rate of cellular processes increases with temperature. By using *in vitro* fertilisation, the timing of fertilisation could be controlled more accurately. When reared at the same standard temperature, diploid and triploid larvae showed similar growth, development, and expression levels for six housekeeping genes; minor differences in gene expression were transient and confined to the first two days directly following cold shock. Based on the similar expression values of *hsp70.1* in diploids and triploids, it seems that the cold shock did not induce a thermal stress response in triploid larvae.

Triploid and diploid larvae had the same growth trajectories and reached the same length at 5 dpf. As triploid larvae were composed of fewer, yet larger cells, triploids should have a lower rate of cell division. Although the G2/G1 ratio is often interpreted as a measure of growth rate, with higher ratios indicating faster rates of cell division, we observed a higher G2/G1 ratio in triploids ($16.27 \pm 5.27$ s.d.) than in diploids ($10.95 \pm 3.17$ s.d.). Possibly, the G2 phase is lengthened in triploids and hence we find relatively more cells in the G2 phase in triploids. In this case, the higher ratio in triploids actually indicates a slower rate of cell division. A slower replication of larger genomes has indeed been noted before [62]. Species that have undergone genome enlargements in their recent evolutionary history can compensate for the slower replication with an increased rDNA copy number [63,64]. However, in our study, we induced triploidy artificially in our zebrafish, so there was no compensation in the number of rDNA copy numbers relative to the total genome. This absence of compensatory changes to shorten the cell cycle may explain why there were potentially fewer cell divisions and a higher G2/G1 ratio in triploids.

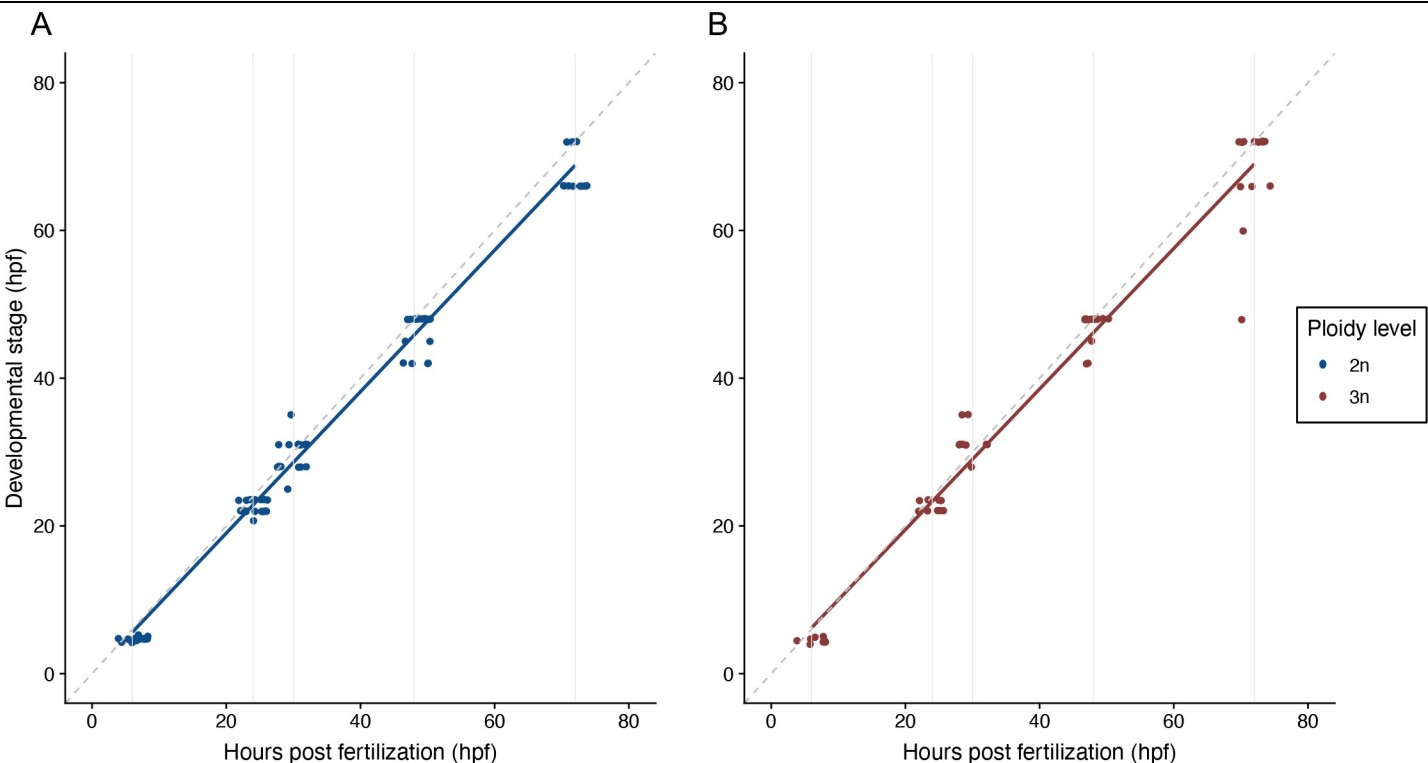

**Fig 7. Development of diploid and triploid larvae up to 72 hpf.** A) Development of diploid larvae, n = 82. B) Development of triploid larvae, n = 55. In both A) and B) the dashed grey line is the reference x = y. Solid grey lines are the hours post fertilization at which the embryos and larvae were staged, namely: 6, 24, 30, 48 and 72 hours. ANOVA, *p* = 0.47.

Interestingly, we observed a small fraction of polyploid cells in both diploid (6n an 8n DNA content) and triploid (9n DNA content) larvae. These cells are originally mononucleated cells, as we measured the DNA content in individual nuclei with flow cytometry. Therefore, it is unlikely that these polyploid cell populations represent muscle cells. While muscle cells have a higher DNA content, the DNA is packaged in multiple nuclei as muscle cells arise from cell fusion of myoblasts [65,66]. Instead, this indicates somatic polyploidy in organogenesis in developing zebrafish. Polyploidization of cells can be a growth strategy to control organ size and morphology [66] and it has been proposed as a way to promote rapid growth, by increasing cell volume without a mitotic division [67].

The observed expression levels of housekeeping genes were very similar for diploid and triploid larvae up until 5 dpf. At first glance, one would predict higher expression values in triploids for all housekeeping genes, as there is simply more DNA available for transcription. However, we need to consider that triploid larvae are made up of fewer cells compared to diploids, and the cDNA used for qPCR is synthesized based on the total amount of RNA. Ribosomal RNA makes up about 50% of the cell's total RNA [68] and the proportions of ribosomal RNA and messenger RNA may change with increasing ploidy. Indeed, gene expression values were lower in triploids for *rpl13a* and *polr2d*, of which the protein products (ribosomal protein L13a and RNA polymerase II subunit D) are involved in the processes of translation and transcription, respectively. Like *polr2d*, *tbp* is important in transcription, coding for the TATA box binding protein [69], but in contrast to *polr2d*, *tbp* is expressed more in triploids, likely to compensate for lower transcription rates and inherently slower cell cycles in triploids.

A larger cell size is hypothesized to have consequences for oxygen transport across the plasma membranes to the mitochondria. Larger cells have relatively less membrane surface to

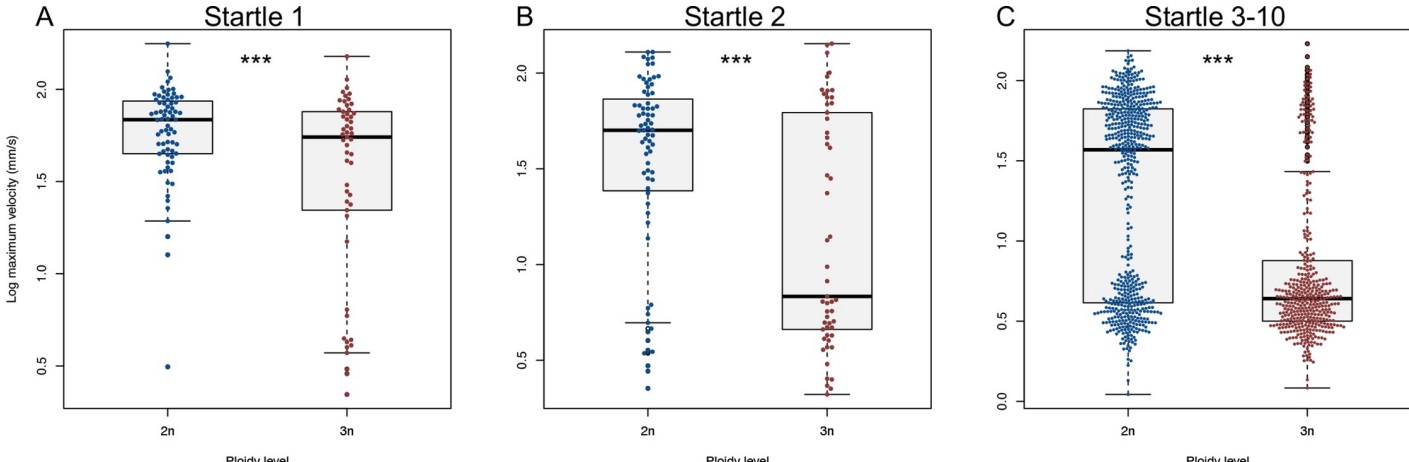

**Fig 8. Startle response of diploid and triploid larvae.** A) Swimming velocity of diploid and triploid larvae at the first startle stimulus. B) Swimming velocity of diploid and triploid larvae at the second startle stimulus. C) Swimming velocity of diploid and triploid larvae at the third to tenth startle stimulus. ANOVA, *** $p < 0.001$, $n = 124$.

transport oxygen into the cells [10,70], which could lower their aerobic energy budget. Temperature changes the balance between oxygen demand and oxygen supply [71,72]. Therefore, triploid zebrafish could be a suitable vertebrate model to study the energetic and respiratory consequences of cell size in thermal biology. Our results on larval swimming performance are consistent with the hypothesis that a triploid larva composed of larger cells has a lower capacity for oxygen provisioning to the mitochondria and might become prone to oxygen shortage upon exercise. The lower maximum swimming velocity in triploids, especially after multiple stimulations suggests that larvae and especially triploids, indeed run out of energy. In adult fish, short burst activity is known to be fuelled predominantly by anaerobic, glycogenic fast muscle metabolism [73,74]. However, El-Fiky et al. (1987) [75] argue that in larval fish jerky and erratic movements are fuelled almost entirely by aerobic metabolism, based on the high activity of aerobic enzymes like cytochrome oxidase and citrate synthase in whole-body and muscle homogenates of young larvae. These markers decrease upon progression through the juvenile stage, and markers for glycolytic enzymes increase [75,76], indicating that the onset of anaerobic power develops later in ontogeny. Although the sensorimotor pathway of the startle response in larval zebrafish is well understood [77,78], the energy source for this response has not been thoroughly investigated. If the energy is predominantly generated aerobically, oxygen limitation could be a reason why triploids show a decreased response after multiple stimuli. Given the all or nothing type of escape response, the lower flux of oxygen in triploids could mean they need longer to 'recharge' before initiating another burst response.

In our study triploid larvae failed to maintain a high swimming activity throughout the trial. It is unlikely that some other defect, such as a sensory blockade or a reduced muscle innervation, is the cause of this, because our analysis only included diploid and triploid larvae that exhibited a startle response. In addition, triploid and diploid larvae did not differ in body length and morphology at 5 dpf, ruling out the possibility that a delayed development caused the reduced swimming performance in triploids. Still, we did not perform a full transcriptome wide analysis and therefore we cannot exclude differential gene expression of genes related to swimming performance (e.g. citrate synthase, lactate dehydrogenases and other metabolic genes).

In summary, our results show that triploidy induction alters cellular architecture and energy metabolism in zebrafish larvae. In other respects, diploid and triploid zebrafish larvae are largely similar, including gene expression at 5 dpf, development, and growth. As these larvae are not dependent yet on gills for breathing or external food, they make for an excellent model to study regulation of cell size and its consequences for animal performance.

## Supporting information

**S1 Table. Triploidy induction efficiency.**
(DOCX)

**S1 Appendix. Analysis of responders per startle.**
(DOCX)

## Acknowledgments

We would like to thank professor H. Komen from Wageningen University & Research (department of Animal Sciences) for the suggestion to use cold shocks to induce triploidy in zebrafish. We are also thankful to Mr. T. Spanings for husbandry of the zebrafish in our fish facility. For the experimental work, we are grateful to Mr. J. Zethof who assisted meticulously in molecular work. We would also like to thank Mr. R. Woestenenk from the Radboud University Medical Center for his help in operating the flow cytometer, and Dr. T. Smith from Carleton University and Mr. P. Kron from the University of Guelph, Canada, for their help in analysing and interpreting the flow cytometry data. The DanioVision experiments were facilitated by Dr. E. van Wijk and Dr. E. de Vrieze (both Radboud University Medical Center, department of Otorhinolaryngology), and we would like to thank Dr. R. van den Bos (department of Animal Ecology and Physiology) for his help in analysing DanioVision data. Finally, we are thankful to Dr. A. Hermaniuk (University of Bialystok, Institute of Biology) for his help in staging the embryos.

## Author Contributions

**Conceptualization:** Iris L. E. van de Pol, Wilco C. E. P. Verberk.

**Funding acquisition:** Wilco C. E. P. Verberk.

**Investigation:** Iris L. E. van de Pol.

**Methodology:** Iris L. E. van de Pol, Wilco C. E. P. Verberk.

**Supervision:** Wilco C. E. P. Verberk.

**Visualization:** Iris L. E. van de Pol.

**Writing – original draft:** Iris L. E. van de Pol.

**Writing – review & editing:** Iris L. E. van de Pol, Gert Flik, Wilco C. E. P. Verberk.

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
