## [Decision Letter · Decision Letter 0]

9 Dec 2019

PONE-D-19-30794

Triploidy in zebrafish larvae: effects on gene expression, cell size and number, growth, development and swimming performance.

PLOS ONE

Dear Ms. van de Pol,

Thank you for submitting your manuscript to PLOS ONE. After careful consideration, we feel that it has merit but does not fully meet PLOS ONE’s publication criteria as it currently stands. Therefore, we invite you to submit a revised version of the manuscript that addresses the points raised during the review process.

Both reviewers found the study interesting with potential to provide a valuable system for testing effects of cell size on physiology. The only major concern, mentioned by both reviewers, is the lack of direct evidence to support the key conclusion of a difference in cells size. It will be important to provide a direct measurement of cell size as part of a revised submission. I also strongly recommend that you provide additional evidence that the FACs method accurately assesses ploidy as the conclusions rely heavily on the successful induction of triploidy. The reviewers raised a number of other points that can be addressed by clarifying the text, adding caveats or making minor corrections.

Finally, I was also fascinated by this study and have two minor comments for you to optionally consider. (1) Is it possible that the fraction of cells in normal zebrafish that are polyploid are simply muscle cells? If so, perhaps its worth mentioning briefly. (2) Fig 7 is not entirely persuasive regarding greater fatigue, rather than habituation, which may not refelect an energy deficit. Based on the threshold of 15 mm/s mentioned in the Methods, it appears that Fig 7 shows both responders and non-responders. Perhaps I am mistaken but it appears that with repeated trials the triploids tend to become non-responders (habituation) rather than show a decrease in velocity (fatigue) - or perhaps both effects are present. As such it would be useful to provide a figure that contrasted the mean/SEM only for responders.

We would appreciate receiving your revised manuscript by Jan 23 2020 11:59PM. To enhance the reproducibility of your results, we recommend that if applicable you deposit your laboratory protocols in protocols.io, where a protocol can be assigned its own identifier (DOI) such that it can be cited independently in the future. For instructions see: http://journals.plos.org/plosone/s/submission-guidelines#loc-laboratory-protocols

We look forward to receiving your revised manuscript.

Kind regards,

Harold A. Burgess, Ph.D.

Academic Editor

PLOS ONE

Please ensure that your manuscript meets PLOS ONE's style requirements, including those for file naming. The PLOS ONE style templates can be found at http://www.plosone.org/attachments/PLOSOne_formatting_sample_main_body.pdf and http://www.plosone.org/attachments/PLOSOne_formatting_sample_title_authors_affiliations.pdf

Reviewers' comments:

Reviewer's Responses to Questions

**Comments to the Author**

1. Is the manuscript technically sound, and do the data support the conclusions?

Reviewer #1: Partly

Reviewer #2: Partly

2. Has the statistical analysis been performed appropriately and rigorously? 

Reviewer #1: Yes

Reviewer #2: I Don't Know

3. Have the authors made all data underlying the findings in their manuscript fully available?

Reviewer #1: Yes

Reviewer #2: Yes

4. Is the manuscript presented in an intelligible fashion and written in standard English?

Reviewer #1: Yes

Reviewer #2: No

5. Review Comments to the Author

Reviewer #1: This is an original study that provides new information characterizing triploid vs diploid zebrafish during the larval period. This is useful information that is generally appropriately collected and reported. While the authors do not observe changes in body size or expression of housekeeping genes between diploids and triploids, they observe reduced cell numbers in otherwise normal appearing zebrafish and a diminished startle response.

My main concern is that some of the data may be over interpreted, and that the manuscript may be improved by focusing on the differences observed between diploids and triploids, reserving implied conclusions for the discussion. For example

The authors don’t directly measure cell size, so without additional data, this is an inference that shouldn’t really be stated as a fact in the title. The authors report of a modest but significant reduction in cell number in triploids compared to diploids without a change in body length is consistent with their inference. Therefore, it is reasonable to state this in the text. However, weight isn’t reported, which could significantly impact this inference.

There are also likely to be gene expression changes in triploid vs diploid larvae. Without RNAseq to rule out this possibility, the authors should be more careful about

attributing phenotypes to cell size and they should acknowledge this caveat more strongly when discussing triploid zebrafish as a model of increased cell size.

Additional minor comments:

For better transparency, length measurements should ideally be presented in a way that includes individual data points rather than the aggregate data in a bar graph

Scale bars are needed on larval images

The writing in the section below is confusing. The first sentence seems to say triploid zebrafish die around 50 days, while the last suggests they live up to a year?

Ploidy can be artificially increased in zebrafish, either by inducing triploidy [29-31], or tetraploidy, although these fish do not to survive beyond 50 days of age [32]. This is surprising in the light of the tetraploidy of the closely related common carp (Cyprinus carpio). Triploid zebrafish, however, survive easily well up to adult age (> 1 year, personal

110 observations), although they all develop into males [33].

Reviewer #2: Review: Triploidy in zebrafish larvae: effects on gene expression, cell size and number, growth, development and swimming performance

The manuscript describes the use of triploid zebrafish as a model to study the consequences of cell size. The manuscript is interesting. However, it is missing some key experiments to support the conclusions drawn. Mainly, these zebrafish are to be used as a model to study the consequences of cell size adaptation on energy; however cell size and the effect of cell size on energy was not directly analyzed. There were also some inconsistencies in the data that should be clarified (please see comments below). Furthermore, the manuscript should be revised to correct the grammatical errors, errors in gene names, and clarify some statements (please see comments below).

Comments:

1. Please provide the full gene names when first used and please use the correct and current gene names. For example ef1a should be (eef1a1l1).

2. Line 53-65: The introduction talks about the effect of temperature on cell size. Although the authors rear their fish at a consistent temperature, they did not test the effect of temperature on cell size. It is suggested that the authors remove this from the text or add in some data regarding the effect of temperature on cell size in diploid and triploid cell size in zebrafish.

3. Line 193 – In the S1 Table triploidy induction at 41 degrees was performed using natural fertilization. Could the authors provide data using ivf or indicate a consistent timepoint used to determine fertilization? Furthermore, considering the authors compare the efficiency of their method to other conflicting published methods, they should mention the effect of potential timing irregularities and the potential effects of using natural fertilization as opposed to ivf with regards to the efficiency of their results.

4. Line 366-381 – The authors confirm triploidy through flow cytometry, with triploids showing a shift in peaks so that a new peak is formed between the G1 and G2 peaks. While I agree the data likely confirms the induction of triploidy, it is suggested that the authors add references to past publications that successfully use this method to confirm triploidy or further discuss the accuracy of the additional peak between G1 and G2 in determining triploidy in zebrafish. Alternatively, it is suggested that the authors confirm the accuracy of this method through the addition of previous published methods (i.e. Giemsa stain on blood smears). Furthermore, on line 377 - the authors state they expected “a factor of 1.5”. Could the authors provide more details/clarify why they expected a factor of 1.5 or provide a reference.

5. Line 376 – The authors state that the average cell count is 1.72 times lower in triploids compared to diploids and that since the diploid and triploid larvae were of the same cell length, triploid larvae can be “inferred to consist of larger but fewer cells compared to the diploid counterparts”. Considering the authors are suggesting the use of these triploids as “a model system to test for the consequences of cell size adaptation”, it is suggested that the authors provide quantitative data demonstrating the size differences between the diploids and triploids. For example: the authors could perform immunohistochemistry for a cell membrane marker followed by confocal microscopy to measure cell size at consistent locations on the embryo/larvae.

6. Line 390 – the authors state that there were “similar patterns of expression for the other three genes”. Could the authors please reword to remove “patterns of expression” when referring to qRT-PCR data as it infers expression in particular regions/tissues (i.e. in situ hybridizations). Furthermore, could the authors specify which “three genes” they are referring too as there are significant differences in the graphs on Figure 3.

7. Line 391-393: The authors should write the full and correct name of the genes of interest. Furthermore, it would benefit the article to explain why these genes were chosen prior to the discussion.

8. Figure 3: It is suggested that the authors indicate where the significant increase in rpl13 and decrease in tbp is on the graphs as it is shown in the polr2d graph. Furthermore, it is suggested that the authors remove the asterisks in the titles for consistency. Furthermore, it appears as though there may be a significant difference 4dpf for b-actin. Could the authors please clarify?

9. Line 398 – The authors indicate there was no significant difference in hsp70l expression. In the discussion the authors state that the absence of an increase in hsp70l expression indicates a lack of thermal stress. However, could it be that hsp70l in zebrafish is only expressed in response to heat instead of cold? Could the authors provide references that indicate hsp70l is expressed specifically in response to cold induced thermal stress in zebrafish?

10. Line 407 – The authors state that the qRT-PCR was normalized using five housekeeping genes, however, the text states only 2 housekeeping genes were used. Could the authors please clarify. Similarly, in the discussion on line 472 – the authors state the “expression pattern for six housekeeping genes” were similar. Please clarify which genes are being referred to and how they are similar as it is unclear as written.

11. Line 477 – The authors state that the larvae reached the same “body size at 5dpf”. Please change to length as it was the only factor measured.

12. Line 496 – The authors use the term “Similar expression patterns” for qRT-PCR. As mentioned previously, please reword. Furthermore, please reword “throughout their development” as expression was only observed until 5dpf.

13. Line 478/480 – The authors state that “Although the G2/G1….in triploids”. Could the authors provide the G2/G1 ratio for the triploids?

14. Line 482- Should “growth rate” be changed to “rate of cell division”. The provided data indicates that the growth rate between the triploids and diploids is similar. However, the authors indicate the rate of cell division may be slower based on flow cytometry data.

15. Line 488 – Please add “potentially” before fewer cell divisions as that data has not been shown, but inferred.

16. Line 496-508 - the authors suggest that there are a large variety of changes that are happening with regards to ribosomal and messenger RNA. Some examples of these changes include changes in the expression of transcription and translation factors such as rpl13, polr2d and tbp. I agree that these results indicate there are potentially changes in transcription and translation. However, I am not sure that these results indicate that transcription/translation rates are decreased or that tpb is compensating for the decreased transcription/translation as a result of polr2d and rpl13. Could the authors please either clarify/expand this statement or provide further evidence to show changes in transcription rate and compensation by tbp.

17. Line 515 – The authors state that “our results on swimming performance are consistent with the hypothesis that a triploid larva composed of larger cells has a lower capacity for oxygen provisioning to the mitochondria and might become prone to oxygen shortage upon exercise.” Considering the intended use of the model and the above statement the publication would benefit from additional data (for example, a seahorse assay, or FACS followed by ATP assay following the treatments) to confirm their hypothesis with regards to the triploid cells possessing a lower capacity to generate energy.

6. PLOS authors have the option to publish the peer review history of their article (what does this mean?). If published, this will include your full peer review and any attached files.

Reviewer #1: No

Reviewer #2: No

---

## [Author Response · Author response to Decision Letter 0]

20 Jan 2020

Dear Editor,

We thank you for providing us the opportunity to submit a revised version of our manuscript.

We feel that the feedback and constructive advice from both reviewers and yourself helped us to further improve our manuscript. 

We added a direct measurement of cell size on erythrocytes of diploid and triploid zebrafish, showing that cell volumes are increased by a factor of 1.64 (± 0.18 s.d.) in triploid zebrafish, which closely agrees with the predicted factor of 1.5 based on their larger genome. We also explained in more detail the flow cytometry method we use to determine ploidy levels of zebrafish larvae, which is based on standard practice protocols (references have now been added) for assessment of ploidy levels in different organisms. Lastly, we corrected grammatical inconsistencies and reworded some phrases to clarify our findings. 

We feel we were able to address and incorporate the comments and suggestions made by the reviewers and yourself. A detailed description of the specific changes made can be found below.

Best wishes, also on behalf of the co-authors,

Iris van de Pol

ACADEMIC EDITOR COMMENTS TO THE AUTHOR

Academic Editor

Comments to the author:

Both reviewers found the study interesting with potential to provide a valuable system for testing effects of cell size on physiology. The only major concern, mentioned by both reviewers, is the lack of direct evidence to support the key conclusion of a difference in cells size. It will be important to provide a direct measurement of cell size as part of a revised submission. The reviewers raised a number of other points that can be addressed by clarifying the text, adding caveats or making minor corrections. 

Reply: We thank you and the reviewers for your appreciation of our manuscript. We have addressed all concerns of the reviewers, adding a direct measurement of cell size and adding extra evidence to support the validity of our flow cytometry method to assess ploidy levels (see below for a detailed response).

I also strongly recommend that you provide additional evidence that the FACs method accurately assesses ploidy as the conclusions rely heavily on the successful induction of triploidy. 

Reply: We have added appropriate references to inform the reader on our procedure to assess ploidy level using flow cytometry (line 293). Adding an internal control with known ploidy level is a common procedure that makes it possible to relate fluorescent intensities of the peaks in the obtained DNA histograms to the corresponding genome sizes. Although our method is new in its application to zebrafish larvae, the basic idea behind it is not. Therefore, we are confident that triploidy induction can be verified unambiguously with this method.

Finally, I was also fascinated by this study and have two minor comments for you to optionally consider. 

(1) Is it possible that the fraction of cells in normal zebrafish that are polyploid are simply muscle cells? If so, perhaps it’s worth mentioning briefly. 

Reply: We have thought of this, but skeletal muscle cells in vertebrates arise form cell fusion, giving rise to a syncytium; a large multinucleated cell. Hence, the individual nuclei of such muscle cells are still of the same ploidy. As we measured DNA content in individual nuclei, our finding of polyploid nuclei cannot be nuclei derived from multinucleated muscle cells. We have added this consideration in the Discussion section (lines 543-545).

(2) Fig 7 is not entirely persuasive regarding greater fatigue, rather than habituation, which may not reflect an energy deficit. Based on the threshold of 15 mm/s mentioned in the Methods, it appears that Fig 7 shows both responders and non-responders. Perhaps I am mistaken but it appears that with repeated trials the triploids tend to become non-responders (habituation) rather than show a decrease in velocity (fatigue) - or perhaps both effects are present. As such it would be useful to provide a figure that contrasted the mean/SEM only for responders.

Reply: We agree that it is difficult to distinguish between fatigue and habituation. However, we have chosen this startle protocol with a 20 s interval between stimuli as previous studies showed that no habituation takes place with this time interval. We have added this information including references in the M&M section (lines 349-351). Therefore, it is unlikely that habituation is the cause of the lower responsiveness of triploids after multiple stimuli.

With regard to Fig 7, we clarified our definition of responders and non-responders in the M&M section. Fig 7 only includes responders, defined as larvae that at least once show a response higher than 15 mm/s during the startle protocol. This means they are physically able to respond to a stimulus, although they might not have responded to all 10 stimuli. Thus, Fig 7 also included larvae that did not respond to a given stimulus. There is indeed a bimodality in responses: larvae either respond to a stimulus or they do not. Therefore, we performed a χ2 test to determine the proportion of larvae that responded to a stimulus per startle, and we calculated the average maximum swimming velocity of the larvae that responded to a stimulus. Although the number of triploids that responded to a stimulus was lower for all startles, when they did respond their maximum swimming velocity did not differ from that of diploids. This indicates an all or nothing escape response, and the decreased responsiveness of triploids after subsequent stimuli is driving this pattern. As mentioned above, habituation is unlikely as a cause, but given the all-or nothing escape response, triploid larvae could need more time to ‘recharge’ to initiate another burst activity response. We have added the new results of the χ2 test and clarified our interpretation of the results (lines 479-485).

REVIEWER’S COMMENTS TO THE AUTHOR

Reviewer 1

Comments to the author:

The authors don’t directly measure cell size, so without additional data, this is an inference that shouldn’t really be stated as a fact in the title. The authors report of a modest but significant reduction in cell number in triploids compared to diploids without a change in body length is consistent with their inference. Therefore, it is reasonable to state this in the text. However, weight isn’t reported, which could significantly impact this inference.

Reply: We agree that a direct measurement of cells size strengthens our conclusion that triploid larvae are made up of fewer, but larger cells. Therefore, we added a measurement of cell size of erythrocytes of adult diploid and triploid zebrafish, showing that triploidy induction indeed leads to larger cells. With regard to reporting weight; we have tried to determine dry mass of individual larvae but the results were not reliable as their weight is very low (<0.1 mg). We would also like to point out that both diploid and triploid larvae arise from eggs which were harvested from the same population of (diploid) females. Thus, egg mass and yolk content were similar at the start and as the larvae do not feed up until 5 dpf, this provides an additional reason why it is unlikely that diploids and triploids would differ in weight. 

There are also likely to be gene expression changes in triploid vs diploid larvae. Without RNAseq to rule out this possibility, the authors should be more careful about

attributing phenotypes to cell size and they should acknowledge this caveat more strongly when discussing triploid zebrafish as a model of increased cell size.

Reply: We followed the suggestion of the reviewer and added a caveat in the discussion where we included a statement acknowledging the fact that we did not analyse the entire transcriptome (lines 590-593).

Additional minor comments:

For better transparency, length measurements should ideally be presented in a way that includes individual data points rather than the aggregate data in a bar graph

Reply: Done.

Scale bars are needed on larval images

Reply: Done.

The writing in the section below is confusing. The first sentence seems to say triploid zebrafish die around 50 days, while the last suggests they live up to a year?

Ploidy can be artificially increased in zebrafish, either by inducing triploidy [29-31], or tetraploidy, although these fish do not to survive beyond 50 days of age [32]. This is surprising in the light of the tetraploidy of the closely related common carp (Cyprinus carpio). Triploid zebrafish, however, survive easily well up to adult age (> 1 year, personal

110 observations), although they all develop into males [33].

Reply: We apologise for the confusion and have now clarified in the text that we meant only tetraploid zebrafish have been reported to live up to 50 days (line 108). 

Reviewer 2

Comments to the author:

The manuscript describes the use of triploid zebrafish as a model to study the consequences of cell size. The manuscript is interesting. However, it is missing some key experiments to support the conclusions drawn. Mainly, these zebrafish are to be used as a model to study the consequences of cell size adaptation on energy; however, cell size and the effect of cell size on energy was not directly analyzed.

Reply: Indeed, we did not measure cell size directly in the first version of our manuscript. As stated above, we agree that direct measurements of cell size strengthen our study. Therefore, we added measurements of erythrocytes from adult diploid and triploid zebrafish. The results show that triploidy induction is indeed followed by an increase in cell size.

The consequences of enlarged cells for energy budgeting in zebrafish larvae remain to be elucidated. We have shown a greater fatigue in triploid larvae after repeated stimuli in a startle protocol, which aligns with the hypothesis that larger cells result in a lower capacity for performance. In follow-up studies we will address the physiological mechanisms behind the observed response of a decreased swimming performance in triploids.

1. Please provide the full gene names when first used and please use the correct and current gene names. For example ef1a should be (eef1a1l1).

Reply: Done.

2. Line 53-65: The introduction talks about the effect of temperature on cell size. Although the authors rear their fish at a consistent temperature, they did not test the effect of temperature on cell size. It is suggested that the authors remove this from the text or add in some data regarding the effect of temperature on cell size in diploid and triploid cell size in zebrafish.

Reply: To the best of our abilities, we provide information in the introduction that supports the relevance of our study. Our primary motivation to create a zebrafish model with larger cells is to eventually use this model to study the role of cell size in mediating temperature and oxygen effects. Such a model does not exist yet, and we feel that the approach we take in the introduction, by starting off from an ecophysiology perspective, justifies the need to create a model to study the consequences of cell size adaptations. As such, we would like to keep this information in the introduction. 

3. Line 193 – In the S1 Table triploidy induction at 41 degrees was performed using natural fertilization. Could the authors provide data using ivf or indicate a consistent timepoint used to determine fertilization? Furthermore, considering the authors compare the efficiency of their method to other conflicting published methods, they should mention the effect of potential timing irregularities and the potential effects of using natural fertilization as opposed to ivf with regards to the efficiency of their results.

Reply: We have now included that we visually assessed the timing of egg release by the female during natural fertilization (S1 Table). We then started a timer, but sometimes small timing irregularities could indeed occur. In the discussion section we added that small timing irregularities probably have larger effects during heat shocks than during cold shocks, as the rate of cellular processes increases with high temperatures (lines 517-520).

4. Line 366-381 – The authors confirm triploidy through flow cytometry, with triploids showing a shift in peaks so that a new peak is formed between the G1 and G2 peaks. While I agree the data likely confirms the induction of triploidy, it is suggested that the authors add references to past publications that successfully use this method to confirm triploidy or further discuss the accuracy of the additional peak between G1 and G2 in determining triploidy in zebrafish. Alternatively, it is suggested that the authors confirm the accuracy of this method through the addition of previous published methods (i.e. Giemsa stain on blood smears). Furthermore, on line 377 - the authors state they expected “a factor of 1.5”. Could the authors provide more details/clarify why they expected a factor of 1.5 or provide a reference.

Reply: We have added the desired reference (line 406). As stated above, it is a standard practice to add an internal control with known ploidy level when using flow cytometry. This is necessary to be able to relate fluorescent intensities of the peaks in the obtained DNA histograms to the corresponding genome sizes.

We predict an increase factor of 1.5 in cell size because the genome size was increased by the same factor and multiple studies (references have been added in line 70) have demonstrated a positive relationship between genome size, nucleus size and cell size.

5. Line 376 – The authors state that the average cell count is 1.72 times lower in triploids compared to diploids and that since the diploid and triploid larvae were of the same cell length, triploid larvae can be “inferred to consist of larger but fewer cells compared to the diploid counterparts”. Considering the authors are suggesting the use of these triploids as “a model system to test for the consequences of cell size adaptation”, it is suggested that the authors provide quantitative data demonstrating the size differences between the diploids and triploids. For example: the authors could perform immunohistochemistry for a cell membrane marker followed by confocal microscopy to measure cell size at consistent locations on the embryo/larvae.

Reply: We agree that direct measurements of cell size in larvae would be ideal. We have tried to stain cell membranes in whole larvae and image them with a confocal microscope. However, we had some trouble with the penetration of the stain into the larvae; some parts were overstained, while others (deeper in the larvae) were not stained at all. It was also difficult to standardize a region to measure cell size between larvae, as specific organ structures are hard to identify. Therefore, we chose to measure cell size in blood smears form adult diploid and triploid zebrafish. 

6. Line 390 – the authors state that there were “similar patterns of expression for the other three genes”. Could the authors please reword to remove “patterns of expression” when referring to qRT-PCR data as it infers expression in particular regions/tissues (i.e. in situ hybridizations). Furthermore, could the authors specify which “three genes” they are referring too as there are significant differences in the graphs on Figure 3.

Reply: We have reworded “patterns of expression” to “levels of expression” throughout the manuscript. Indeed, there are significant differences in three genes (rpl13a, tbp and polr2d) but the trajectory of ontogenetic changes in expression levels are very similar for these genes. We have now rephrased this in the manuscript (line 427).

7. Line 391-393: The authors should write the full and correct name of the genes of interest. Furthermore, it would benefit the article to explain why these genes were chosen prior to the discussion.

Reply: We followed this advice and now provide the full names for the chosen genes in the methods section (lines 248-252). We also explain that the reason for choosing these genes is because they are involved in a range of cellular processes, and therefore they are also frequently used to normalize gene expression levels of other genes (lines 252-253).

8. Figure 3: It is suggested that the authors indicate where the significant increase in rpl13 and decrease in tbp is on the graphs as it is shown in the polr2d graph. Furthermore, it is suggested that the authors remove the asterisks in the titles for consistency. Furthermore, it appears as though there may be a significant difference 4dpf for b-actin. Could the authors please clarify?

Reply: For rpl13a and tbp there was a significant difference in gene expression between diploids and triploids, but the interaction with dpf was not significant. Therefore, we did not include the asterisks in the graph as for polr2d. We clarified this in the figure legend. We agree that actb1 levels appear to be different at 4 dpf, but in the overall analysis there was no significant interaction and therefore these additional posthoc tests were not warranted. Note that when we still (inappropriately) contrasted diploids and triploids there was a significant difference at day 4 (p=0.014).

9. Line 398 – The authors indicate there was no significant difference in hsp70l expression. In the discussion the authors state that the absence of an increase in hsp70l expression indicates a lack of thermal stress. However, could it be that hsp70l in zebrafish is only expressed in response to heat instead of cold? Could the authors provide references that indicate hsp70l is expressed specifically in response to cold induced thermal stress in zebrafish?

Reply: We added references that show that hsp70.1 can be activated upon cold stress in zebrafish (line 258).

10. Line 407 – The authors state that the qRT-PCR was normalized using five housekeeping genes, however, the text states only 2 housekeeping genes were used. Could the authors please clarify. Similarly, in the discussion on line 472 – the authors state the “expression pattern for six housekeeping genes” were similar. Please clarify which genes are being referred to and how they are similar as it is unclear as written.

Reply: The expression levels of the six housekeeping genes tested were indeed normalized for each gene by using the relative quantity of the other five housekeeping genes. Two housekeeping genes (eef1a1l1 and actb1) were only used to normalize the expression of hsp70.1, as these were most stably expressed during development and similar for diploids and triploids. We now clarified this in the text (lines 436-438).

11. Line 477 – The authors state that the larvae reached the same “body size at 5dpf”. Please change to length as it was the only factor measured.

Reply: Done.

12. Line 496 – The authors use the term “Similar expression patterns” for qRT-PCR. As mentioned previously, please reword. Furthermore, please reword “throughout their development” as expression was only observed until 5dpf.

Reply: Done.

13. Line 478/480 – The authors state that “Although the G2/G1….in triploids”. Could the authors provide the G2/G1 ratio for the triploids?

Reply: Done.

14. Line 482- Should “growth rate” be changed to “rate of cell division”. The provided data indicates that the growth rate between the triploids and diploids is similar. However, the authors indicate the rate of cell division may be slower based on flow cytometry data.

Reply: Thank you for this suggestion. Indeed, rate of cell division is a better way to put it.

15. Line 488 – Please add “potentially” before fewer cell divisions as that data has not been shown, but inferred.

Reply: Done.

16. Line 496-508 - the authors suggest that there are a large variety of changes that are happening with regards to ribosomal and messenger RNA. Some examples of these changes include changes in the expression of transcription and translation factors such as rpl13, polr2d and tbp. I agree that these results indicate there are potentially changes in transcription and translation. However, I am not sure that these results indicate that transcription/translation rates are decreased or that tpb is compensating for the decreased transcription/translation as a result of polr2d and rpl13. Could the authors please either clarify/expand this statement or provide further evidence to show changes in transcription rate and compensation by tbp.

Reply: We agree that our data do not conclusively demonstrate changes in transcription rate and compensation by tbp. We did observe differences in the G1/G2 ratio, suggesting that the G2 phase in lengthened in triploids, and these are consistent with the observed differences in gene expression. We have tempered our wording (lines 530-539).

17. Line 515 – The authors state that “our results on swimming performance are consistent with the hypothesis that a triploid larva composed of larger cells has a lower capacity for oxygen provisioning to the mitochondria and might become prone to oxygen shortage upon exercise.” Considering the intended use of the model and the above statement the publication would benefit from additional data (for example, a seahorse assay, or FACS followed by ATP assay following the treatments) to confirm their hypothesis with regards to the triploid cells possessing a lower capacity to generate energy.

Reply: We agree that an exciting next step is to test the hypothesis that triploid cells have a lower capacity to generate energy. Such differences in capacity to generate energy are likely temperature dependent and therefore rigorous testing needs to study differences between diploids and triploids at different temperatures. We are pursuing this in our current work, but consider these beyond the aim of the present manuscript which is to present the diploid/triploid zebrafish larvae model system, noting the striking similarities under standard temperatures and under non-demanding conditions.

---

## [Decision Letter · Decision Letter 1]

4 Feb 2020

PONE-D-19-30794R1

Triploidy in zebrafish larvae: effects on gene expression, cell size and cell number, growth, development and swimming performance.

PLOS ONE

Dear Ms. van de Pol,

Thank you for submitting your manuscript to PLOS ONE. After careful consideration, we feel that it has merit but does not fully meet PLOS ONE’s publication criteria as it currently stands. Therefore, we invite you to submit a revised version of the manuscript that addresses points raised during the review process.

The reviewer requested that the caveat attached to Table S1 should really appear in the main text somewhere. I think it would be fine to just move this text into the relevant part of the Discussion. This is a very minor point, but as some readers may never actually look a the supplemental table I agree its probably a good idea.

We would appreciate receiving your revised manuscript by Mar 20 2020 11:59PM. To enhance the reproducibility of your results, we recommend that if applicable you deposit your laboratory protocols in protocols.io, where a protocol can be assigned its own identifier (DOI) such that it can be cited independently in the future. For instructions see: http://journals.plos.org/plosone/s/submission-guidelines#loc-laboratory-protocols

We look forward to receiving your revised manuscript.

Kind regards,

Harold A. Burgess, Ph.D.

Academic Editor

PLOS ONE

Reviewers' comments:

Reviewer's Responses to Questions

**Comments to the Author**

1. If the authors have adequately addressed your comments raised in a previous round of review and you feel that this manuscript is now acceptable for publication, you may indicate that here to bypass the “Comments to the Author” section, enter your conflict of interest statement in the “Confidential to Editor” section, and submit your "Accept" recommendation.

Reviewer #1: All comments have been addressed

Reviewer #2: (No Response)

2. Is the manuscript technically sound, and do the data support the conclusions?

Reviewer #1: (No Response)

Reviewer #2: Yes

3. Has the statistical analysis been performed appropriately and rigorously? 

Reviewer #1: (No Response)

Reviewer #2: I Don't Know

4. Have the authors made all data underlying the findings in their manuscript fully available?

Reviewer #1: (No Response)

Reviewer #2: Yes

5. Is the manuscript presented in an intelligible fashion and written in standard English?

Reviewer #1: (No Response)

Reviewer #2: Yes

6. Review Comments to the Author

Reviewer #1: (No Response)

Reviewer #2: The authors have satisfyingly responded to my specific comments, except for

one which is listed below and can be addressed through clarification in the text.

Table S1 – In response to a previous comment (comment #3) the authors added a disclaimer at the end of Table S1 that the timing of fertilization was determined by visually assessing egg release of females during natural fertilization and that this may result in timing irregularities. However, in the body of the text the authors suggest only that in “these studies slightly different conditions were applied”. Considering the authors compare the efficiency of their method to others for heat shock, it should be clearly mentioned in the text that natural fertilization was used for the heat shock studies and that this may result in some timing irregularities, which as they mention strongly affects the success of triploidy induction.

7. PLOS authors have the option to publish the peer review history of their article (what does this mean?). If published, this will include your full peer review and any attached files.

Reviewer #1: No

Reviewer #2: No

---

## [Author Response · Author response to Decision Letter 1]

5 Feb 2020

ACADEMIC EDITOR COMMENTS TO THE AUTHOR

Academic Editor

Comments to the author:

The reviewer requested that the caveat attached to Table S1 should really appear in the main text somewhere. I think it would be fine to just move this text into the relevant part of the Discussion. This is a very minor point, but as some readers may never actually look a the supplemental table I agree its probably a good idea.

Reply: We thank you for this suggestion. We have now included this information in the Discussion section of the manuscript (line 518-523).

REVIEWER’S COMMENTS TO THE AUTHOR

Reviewer 1

Comments to the author:

All comments have been addressed.

Reviewer 2

Comments to the author:

The authors have satisfyingly responded to my specific comments, except for

one which is listed below and can be addressed through clarification in the text.

Table S1 – In response to a previous comment (comment #3) the authors added a disclaimer at the end of Table S1 that the timing of fertilization was determined by visually assessing egg release of females during natural fertilization and that this may result in timing irregularities. However, in the body of the text the authors suggest only that in “these studies slightly different conditions were applied”. Considering the authors compare the efficiency of their method to others for heat shock, it should be clearly mentioned in the text that natural fertilization was used for the heat shock studies and that this may result in some timing irregularities, which as they mention strongly affects the success of triploidy induction.

Reply: Indeed, we agree we should mention this caveat in the main text. Therefore, we included this information to the Discussion section of the manuscript (line 518-523).

---

## [Editor Report · Decision Letter 2]

7 Feb 2020

Triploidy in zebrafish larvae: effects on gene expression, cell size and cell number, growth, development and swimming performance.

PONE-D-19-30794R2

Dear Dr. van de Pol,

We are pleased to inform you that your manuscript has been judged scientifically suitable for publication and will be formally accepted for publication once it complies with all outstanding technical requirements.

With kind regards,

Harold A. Burgess, Ph.D.

Academic Editor

PLOS ONE
---

## [Editor Report · Acceptance letter]

18 Feb 2020

PONE-D-19-30794R2 

Triploidy in zebrafish larvae: effects on gene expression, cell size and cell number, growth, development and swimming performance. 

Dear Dr. van de Pol:

I am pleased to inform you that your manuscript has been deemed suitable for publication in PLOS ONE. Congratulations! Your manuscript is now with our production department. 

With kind regards,

on behalf of

Dr Harold A. Burgess 

Academic Editor

PLOS ONE